Real-world experience of switching from tenofovir disoproxil fumarate to tenofovir alafenamide in patients with chronic hepatitis B: a retrospective study

Su Pei-Yuan 1 111252@cch.org.tw
http://orcid.org/0000-0001-5554-5938 Su Wei-Wen 1
Hsu Yu-Chun 1
http://orcid.org/0000-0003-1021-2114 Huang Siou-Ping 1
http://orcid.org/0000-0002-3494-2245 Yen Hsu-Heng 1 2 3
1 Department of Gastroenterology, Changhua Christian Hospital , Changhua , Taiwan
2 Institute of Medicine, Chung Shan Medical University , Taichung , Taiwan
3 General Education Center, Chienkuo Technology University , Changhua , Taiwan
Sotelo-Mundo Rogerio
Electronic publication date: 2021 Nov 19
Publication date: 2021
Volume: 9
Electronic Location ID: e12527
Received 2021 Aug 18; Accepted 2021 Nov 1
Copyright: © 2021 Su et al.
Copyright year: 2021
Copyright holder: Su et al.
License: This is an open access article distributed under the terms of the Creative Commons Attribution License, which permits unrestricted use, distribution, reproduction and adaptation in any medium and for any purpose provided that it is properly attributed. For attribution, the original author(s), title, publication source (PeerJ) and either DOI or URL of the article must be cited.
License URL: https://creativecommons.org/licenses/by/4.0/

Keywords: Tenofovir Disoproxil Fumarate, Tenofovir alafenamide, Chronic hepatitis B, Switch

Funding: Changhua Christian Hospital 110-CCH-IRP-040 This research was funded by Changhua Christian Hospital (110-CCH-IRP-040). The funders had no role in study design, data collection and analysis, decision to publish, or preparation of the manuscript.

==============================
Background

Tenofovir alafenamide (TAF) has good viral suppression efficacy and less adverse effect than tenofovir disoproxil fumarate (TDF). Real-world studies on the antiviral efficacy and safety of switching from TDF to TAF in patients with chronic hepatitis B (CHB) are limited.

Methods

This retrospective study included 167 nucleos(t)ide analogue (NA)-naive patients with CHB. All the patients received TDF at least 12 months before switching and TAF at least 12 months after switching at a single medical center. The Friedman test with Dunn–Bonferroni post hoc tests and repeated-measures analysis of variance was used to analyze the effect of complete viral suppression, alanine aminotransferase (ALT) level normalization, renal function changes, body weight, and body mass index in the periods before and after switching.

Results

The mean age and TDF treatment duration were 52 ± 11 years and 2.8 years (interquartile range, 1.51–5.15 years), respectively. The complete viral suppression rate was similar between the time of switching and 48 weeks after switching to TAF (77.8% vs 76%, P = 1.000). The percentage of alanine aminotransferase (ALT) normalization increased from 26.3% at TDF start to 81.4% (P < 0.001) at time of switching and 89.2% at 48 weeks after switching to TAF (P = 0.428). The median estimated glomerular filtration rate decreased from 100.09 mL/min/1.73 m² at TDF start to 91.97 mL/min/1.73 m² (P < 0.001) at the time of switching and stabilized at 48 weeks after switching to TAF (93.47 mL/min/1.73m², P = 1.000). The body weight decreased from 69.2 ± 12.2 kg at TDF start to 67.4 ± 12.1 kg (P < 0.001) at the time of switching to TAF and returned to 68.7 ± 12.7 kg (P < 0.001) 48 weeks thereafter. The body mass index (BMI) decreased from 25 ± 3.3 kg/m² at TDF start to 24.5 ± 3.3 kg/m² (P = 0.002) at the time of switching to TAF and returned to 25.1 ± 3.6 kg/m² (P < 0.001) 48 weeks thereafter.

Conclusions

Our study showed that switching to TAF from TDF had good antiviral effectiveness and stabilized renal function. The body weight and BMI decreased during TDF therapy and regained after switching to TAF.

Introduction

Chronic hepatitis B (CHB) virus (HBV) infection has an incidence rate of 3.5% in the global population according to the World Health Organization estimate in 2015 and the highest incidence rates in the Western Pacific (6.2%) and African regions (6.1%) (WHO, 2017; Yuen et al., 2018). Untreated CHB would lead to cirrhotic complications, liver failure, and/or HCC (Tang et al., 2018). Antiviral therapies with nucleos(t)ide analogues (NA) such as entecavir, tenofovir disoproxil fumarate (TDF), and tenofovir alafenamide (TAF) have been suggested to suppress HBV virus replication and decrease long-term complications and mortality by published guidelines (Lampertico et al., 2017; Terrault et al., 2018).

TDF is an antiviral agent with high potency and genetic barrier. However, its long-term adverse effects on renal function and bone mineral density suggest that older patients or patients with deteriorating renal function and/or osteoporosis should use ETV or TAF rather than TDF (Lampertico et al., 2017). TAF is a prodrug of tenofovir and had a similar antiviral effect on HBV. Two double-blind studies showed that TAF had a high viral suppression rate and less severe bone and renal side effects (Buti et al., 2016; Chan et al., 2016). Recent studies have shown weight increases after switching from TDF to TAF in people living with human immunodeficiency virus (HIV; PLWH) (Schafer et al., 2019; Surial et al., 2021). However, only few studies are available about the real-world data on switching from TDF to TAF in patients with CHB. The purpose of this study was to compare the antiviral effect and changes in renal function, weight, and body mass index (BMI) before and after switching from TDF to TAF in a real-world setting.

Patients and methods

Study population

This retrospective study included NA-naive patients with CHB who received TDF at least 12 months before switching and TAF at least 12 months after switching at a single medical center. All patients had positive hepatitis B surface antigen (HBsAg) for >6 months. Indications for TDF treatment comprised the following: (1) patients with positive HBeAg having ALT levels of ≥5 times the upper limit of normal (ULN); (2) positive HBeAg having HBV deoxyribonucleic acid (DNA) levels of >20,000 IU/ml and ALT levels of >2 times the ULN; (3) negative HBeAg having with HBV DNA levels of >2,000 IU/ml and ALT levels of >2 times the ULN for >3 months; (4) chronic hepatitis B (CHB) having liver decompensation; (5) cirrhosis and HBV DNA levels of >2,000 IU/ml; or (6) CHB receiving chemotherapy (prophylaxis of HBV reactivation). The reasons for switching to TAF were all related to the unavailability of TDF due to the hospital pharmacy policy in our hospital. We excluded patients who had received NA or an interrupted antiviral therapy for >2 weeks, had concomitant medications with other NAs, or were coinfected with hepatitis C virus (HCV) or HIV. The participants were enrolled between December 2011 and March 2021.

Outcomes

The primary outcome was the effect of complete viral suppression over the whole period, defined as a HBV DNA level < 10 IU/mL and normalization of alanine aminotransferase (ALT) level (<40 IU/mL) at TDF start, at switch from TDF to TAF, and after 48 weeks of TAF. The secondary outcomes were changes in renal function, body weight, and BMI before and after switch from TDF to TAF. We collected clinical data, including HBV viral load, liver and kidney functions, and weight at baseline, the time of switching from TDF to TAF, and 48 weeks after switching to TAF. Baseline was defined as the start of using TDF. The estimated glomerular filtration rate (eGFR) was calculated using the Chronic Kidney Disease Epidemiology Collaboration (CKD-EPI) equation (Levey et al., 2009). Cirrhosis was defined based on ultrasonography or radiological evidence of cirrhosis or Fibrosis-4 (FIB-4) scores ≥6.5 or liver histology. The ethics committee of Changhua Christian Hospital approved the study protocol used in the study (institutional review board approval No. 210202). The requirement for informed consent was waived because of the retrospective nature of the study, and the data for analysis were kept anonymous.

Statistical analysis

Data are expressed as n/N (%), median (interquartile range), or mean ± standard deviation. The distribution of continuous variables was checked using the one-sample Kolmogorov–Smirnov test. The Friedman test with Dunn–Bonferroni post hoc tests or repeated-measures analysis of variance was used to compare continuous variables in the three time periods, as appropriate. Categorical variables were compared in the three periods by using the Cochran Q test. All statistical analyses were performed using SPSS version 22.0 (IBM Corp., Armonk, NY, USA), with two-tailed p values < 0.05 indicating statistical significance.

Results

Baseline characteristics at TDF start

Initially, 2,606 NA-naive patients with CHB treated with TDF were enrolled. We excluded 2,439 patients because they withdrew from the study early before starting TAF, could not be contacted, received TDF or TAF treatment for <12 months, and had data unavailable (HBV viral load or GPT). Finally, 167 patients who switched from TDF to TAF for at least 12 months were included in the study (Fig. 1). The patients’ mean age was 52 ± 11 years, and 120 (71.9%) patients were male. The mean BMI and body weight at baseline were 25 ± 3.3 kg/m² and 69.2 ± 12.2 kg, respectively. Of the patients, 109 (65.3%) had cirrhosis while cirrhosis was based on ultrasonography (n = 88), computed tomography (n = 10), magnetic resonance imaging (n = 3), FIB-4 scores ≥ 6.5 (n = 6), or histology (n = 2). The median TDF treatment duration was 2.8 years (interquartile range (IQR), 1.51–5.15 years). The patients’ baseline characteristics are listed in Table 1.

Figure 1 Flow diagram of the patients included in the study.

Table 1 Characteristics at TDF start of the study cohort.

Characteristics at TDF start	n = 167	
Age, years, mean ± SD	52 ± 11	
Sex, male, n (%)	120 (71.9%)	
Body mass index, kg/m², mean ± SD	25 ± 3.3	
Body weight, kg, mean ± SD	69.2 ± 12.2	
HBV viral load, log10IU/mL, mean ± SD	7.56 ± 10.26	
HBeAg positive, n (%)	28 (19.2%)	
ALT, U/L, median (IQR)	64 (38–143)	
eGFR, mL/min/1.73 m², median (IQR)	100.09 (87.44–107.03)	
Cirrhosis, n (%)	109 (65.3%)	
HCC, n (%)	20 (12.0%)	
Malignancy*, n (%)	5 (3%)	
Years on TDF, years, median (IQR)	2.8 (1.51–5.15)	
Notes:

HBeAg, hepatitis B envelope antigen; ALT, alanine transaminase; SD, standard deviation; IQR, interquartile range; HCC, hepatocellular carcinoma; TDF, tenofovir disoproxil fumarate.

* Malignancy included breast cancer (n = 1), bladder cancer (n = 1), supraglottic cancer (n = 1), lung cancer (n = 1) and leukemia (n = 1).

Virological and biochemical outcomes

The mean HBV viral load at baseline, the time of the switch, and 48 weeks after switching were 7.56 ± 10.26, 1.34 ± 2.31, and 1.18 ± 1.97 log10IU/mL, respectively. The percentage of complete viral suppression (HBV < 10 IU/mL) at baseline was 1.2% and significantly increased to 77.8% at the time of the switch (P < 0.001). The rate was similar between the time of switching and 48 weeks after switching to TAF (77.8% vs 76%, P = 1.000; Fig. 2A). The median (IQR) ALT level (IU/mL) at baseline, the time of switching, and 48 weeks after switching were 64 (38–143), 27 (22–39), and 22 (17–31), respectively. The ALT normalization rate (≤40 IU/mL) increased from 26.3% at baseline to 81.4% (P < 0.001) at the time of switching. The rate slightly increased to 89.2% at 48 weeks after switching to TAF but not statistically significantly (P = 0.428; Fig. 2B).

Figure 2 Effect of viral suppression, alanine aminotransferase normalization (ALT) and body weight.

Percentage of complete viral suppression (HBV DNA level < 10 IU/mL) (A), normalization of ALT level (<40 IU/mL) (B), and changes in body mass index (C) at TDF start, the time of switching to TAF, and 48 weeks after switching to TAF. BMI data are shown as mean ± standard deviation (SD).

The median (IQR) eGFR (mL/min/1.73 m²) decreased from 100.09 (87.44–107.03) at baseline to 91.97 (77.19–102.31) (P < 0.001) at the time of switching and stabilized at 48 weeks after switching to TAF (93.34 (79.83–101.89), P = 0.414). A total of 53 patients had complete data on their Fibrosis-4 (FIB-4) scores at the three time points. The median FIB-4 score decreased from 2.93 (1.64–4.89) at baseline to 1.63 (1.17–3.54) at 48 weeks after switching to TAF (P = 0.008). Other characteristics are listed in Table 2.

Table 2 Outcomes of virology, biochemistry, and weight changes at TDF start, the time of switching to TAF, and 48 weeks after switching to TAF.

Characteristics	Sample size	At TDF start
(TB)	Switching to TAF (T0)	48 weeks after switching (T1)	P value	Pairwise comparisons*	
PB0 value	P01 value	PB1 value	
HBV viral load (log10IU/mL),
mean ± SD	167	7.56 ± 10.26	1.34 ± 2.31	1.18 ± 1.97	<0.001	<0.001	1.000	<0.001	
HBeAg positive, n (%)	146	28 (19.2%)	26 (17.8%)	26 (17.8%)	0.264	--	--	--	
ALT, U/L, median (IQR)	167	64 (38–143)	27 (22–39)	22 (17–31)	<0.001	<0.001	<0.001	<0.001	
eGFR, mL/min/1.73 m², median (IQR)	142	100.09
(87.44–107.03)	91.97
(77.19–102.31)	93.34
(79.83–101.89)	<0.001	<0.001	1.000	<0.001	
Total Bilirubin, mg/dL, median (IQR)	50	0.98 (0.57–1.28)	0.8 (0.6–1.1)	0.8 (0.5–1.1)	0.123	--	--	--	
Platelet count, ×10³/μL, median (IQR)	53	142 (102–194)	160 (111–211)	186 (135–234)	0.010	1.000	0.023	0.030	
FIB-4 score, median (IQR)	53	2.93 (1.64–4.89)	1.93 (1.42–3.82)	1.63 (1.17–3.54)	0.011	0.296	0.522	0.008	
Body weight, kg, mean ± SD	136	69.2 ± 12.2	67.4 ± 12.1	68.7 ± 12.7	<0.001	<0.001	<0.001	0.856	
Notes:

* PB0 value: TB vs T0; P01 value: T0 vs T1; PB1 value: TB vs T1

HBeAg, hepatitis B envelope antigen; ALT, alanine transaminase; SD, standard deviation; IQR:,interquartile range; TDF, tenofovir disoproxil fumarate; TAF, tenofovir alafenamide; FIB-4, Fibrosis 4.

Changes in body weight and BMI

The body weight decreased from 69.2 ± 12.2 kg at baseline to 67.4 ± 12.1 kg (P < 0.001) at the time of switching and returned to 68.7 ± 12.7 kg (P < 0.001) at 48 weeks after switching to TAF. The difference in weight between the baseline and 48 weeks after switching was not statistically different (P = 0.856; Table 2). The BMI was also decreased from 25 ± 3.3 kg/m² at baseline to 24.5 ± 3.3 kg/m² (P = 0.002) at the time of switching and returned to 25.1 ± 3.6 kg/m² (P < 0.001) at 48 weeks after switching to TAF. The difference in BMI between the baseline and 48 weeks after switching was also not statistically significant (P = 0.642; Fig. 2C).

Discussion

In this retrospective study, we demonstrated the good antiviral effect of short-term treatment with TAF and better decrease in ALT level after the treatment than at the time of switching and after switching to TDF. The decreased renal function during the TDF therapy was stabilized after switching to TDF. In addition, this real-world study is also one of the first to demonstrate weight changes in patients with chronic hepatitis B infection.

Two previous double-blind randomized trials demonstrated the equal effectiveness of viral suppression when compared with TDF to TAF (Buti et al., 2016; Chan et al., 2016). Another double-blind randomized study about switching from TDF to TAF showed that the viral suppression efficacy of TAF was non-inferior to that of TDF (Lampertico et al., 2020). Our result also showed a similar result that TDF had good virological response, and the effect was sustained after switching to TAF.

TAF increased the ALT normalization rate according to the American Association for the Study of Liver Diseases (AASLD) criteria in the three randomized trials (Lampertico et al., 2020; Agarwal et al., 2018). Recently, three retrospective studies in patients with CHB showed similar findings at 12 to 18 months after switching to TAF, but one study showed that the change was not significant at 6 months after switching to TAF (Toyoda et al., 2021; Bernstein et al., 2021; Farag et al., 2021; Alghamdi et al., 2020). Our study showed that the decrease in ALT level persisted after switching to TAF, but the ALT normalization rate was similar at the time of switching and 48 weeks after switching. In addition, the FIB-4 index showed persistent improvement during such drug switch. The relationship between TDF/TAF and ALT level may be observed at least 48 weeks after switching, and different normalized criteria would affect these results. The associations of ALT level between TDF and TAF were also observed in PLWH. The Swiss HIV Cohort Study showed significant decreased of −11.8 IU/L per year (95% CI [−17.3 to −6.4]) after replacing TDF with TAF in HIV/HBV coinfected patients (Surial et al., 2020). The possible mechanisms remain unknown and may be related to the viral suppression or TDF-induced liver injury (Kovari et al., 2016). Whether the effect is reversible and the relationship between TDF and ALT level in different populations still need confirmation in further large-scale studies.

Renal safety is a major concern in long-term TDF therapy, especially in older patients or patients with renal impairment (Lampertico et al., 2017). TDF can induce renal tubular dysfunction, but the detailed mechanism is unclear (Sano et al., 2021). TAF is a prodrug of tenofovir (TFV) that has less renal toxicity than TDF. The renal function of patients stabilized at 48 weeks after switching to TAF in a randomized trial and four retrospective studies (Lampertico et al., 2020; Toyoda et al., 2021; Bernstein et al., 2021; Farag et al., 2021; Alghamdi et al., 2020). The renal outcome was similar in our study in that the median eGFR decreased during TDF therapy (100.09 vs 91.97 mL/min/1.73 m²) and stabilized at 48 weeks after switching to TAF (93.47 mL/min/1.73 m²).

Body weight changed after switching from TDF to TAF in two retrospective studies in HIV-positive patients in 2019 (Schafer et al., 2019; Gomez et al., 2019). Two large cohort studies showed a mean weight increase of 1.7 kg at 18 months after switching to TAF in Switzerland (Surial et al., 2021) and early and pronounced weight gain (1.80 to 4.47 kg/year) in the United States (Mallon et al., 2021). Recently, a similar finding was found in patients with CHB. The study by Lee et al. (2021) showed a weight increase from 66.6 ± 12.1 kg to 68.1 ± 12.4 at 72 weeks after switching to TAF in 61 Asian patients with CHB. In a randomized non-inferiority trial in Korea, the TAF group showed a significant increase in body weight as compared with the TDF group (0.71 vs −0.37 kg; P = 0.01) (Byun et al., 2021). A decrease in weight was observed in our study during the TDF phase (69.2 ± 12.2 kg to 67.4 ± 12.1 kg, P < 0.001) and returned to the baseline value at 48 weeks after switching to TAF (67.4 ± 12.1 kg to 68.7 ± 12.7 kg, P < 0.001). A similar result in BMI was also observed in the study. The relationship between weight change and TDF or TAF administration remains unclear and requires investigation in further studies.

The study has some limitations. First, this was a retrospective observational study. The laboratory testing could not be performed at regular intervals, and some parameters had missing values. However, the results could still show the statistically significant differences in viral suppression effect and changes in biochemical parameters. Second, we did not measure the serum lipid profiles, fasting glucose and hemoglobin A1c. Some studies showed worsening lipid levels in PLWH after switching to TAF, and the total cholesterol, low-density lipoprotein, and high-density lipoprotein were higher in TAF monotherapy than in TDF monotherapy (Surial et al., 2021; Byun et al., 2021). Additionally, comorbidities, such as diabetes, cardiovascular disease, and malignancy were not adjusted with renal function or weight. These covariates may influence the change in renal function and weight and the association between the covariates and renal function or weight need further investigation. Third, bone mineral density (BMD) was not obtained in our clinical practice. The effect of TDF on BMD is an important issue especially in patients with osteopenia or osteoporosis (Agarwal et al., 2018; Lee et al., 2021) that require long-term antiviral therapy.

In conclusion, the study demonstrated good efficacy of viral suppression and ALT normalization in patients with CHB who switched from TDF to TAF. The stabilization of renal function was observed at 48 weeks after switching to TAF. In addition, our study revealed weight loss and decreased BMI in the TDF phase and regain of weight and BMI at 48 weeks after switching to TAF. Further investigations are needed to evaluate the long-term effect of the clinical implications of switching to TAF in the future.

Supplemental Information

Supplemental Information 1 Raw data of the study.

Click here for additional data file.

Additional Information and Declarations

Competing Interests

Author Contributions

Ethics

Data Availability

The authors declare that they have no competing interests.

Pei-Yuan Su conceived and designed the experiments, prepared figures and/or tables, and approved the final draft.

Wei-Wen Su performed the experiments, authored or reviewed drafts of the paper, and approved the final draft.

Yu-Chun Hsu performed the experiments, prepared figures and/or tables, and approved the final draft.

Siou-Ping Huang analyzed the data, prepared figures and/or tables, and approved the final draft.

Hsu-Heng Yen conceived and designed the experiments, authored or reviewed drafts of the paper, and approved the final draft.

The following information was supplied relating to ethical approvals (i.e., approving body and any reference numbers):

The ethics committee of Changhua Christian Hospital approved the study protocol used in the study (institutional review board approval No. 210202).

The following information was supplied regarding data availability:

The raw measurements are available in the Supplementary File.

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
