# Peer review of "Real-world experience of switching from tenofovir disoproxil fumarate to tenofovir alafenamide in patients with chronic hepatitis B: a retrospective study"

_PeerJ, doi:10.7717/peerj.12527_

## Round 0.1 · original submission · Minor Revisions

Please be encouraged by the prompt reviews, and we look forward to your revised manuscript.

·

Basic reporting

The article is well-structured and very nicely written. The English language is unambiguous and professional. The introduction provides a well-balanced view of the current state of the literature, and references are adequate.

Suggestions for further improvement:
1. Abstract: The methods section should be expanded to include the definition of main outcomes and the statistical methods used to assess them. In addition, consider reporting TDF treatment duration on lines 29-30 as years (interquartile range) instead of years (range) for consistency with the rest of the manuscript.
2. Abstract lines 36-41: These sentences need to be restructured. In its current form, authors list the values first, and the time points afterwards. Consider reformulating it in this form: “The body weight decreased from xxx at TDF start to xxx at the time of switching to TAF, and returned to xxx 48 weeks thereafter.”
1. The first sentence in the introduction should include main keywords for the reader to know what the study is about. As you start with a statement on HCC, one might think that the impact of tenofovir on HCC will be assessed. As this is not the case, consider starting with the WHO estimates or the consequences of untreated CHB instead.
2. Line 65: Consider replacing “densitometry” with “density”.
3. Line 102: Why is “range (minimum/maximum)” included here? Consider removing.
4. HBV viral load: 5 Million +/- 180 Million doesn’t make much sense. This problem arises most commonly with skewed values. I would suggest to report and analyse all HBV viral load values on the log10 scale (line 125 and table 1).
5. Line 171-174: Authors report on a study from the Swiss HIV Cohort Study where switching from TDF to TAF led to decreases in ALT. However, as authors noted, CHB patients were excluded in this study. Consider replacing the reference with Surial B, Béguelin C, Chave J-P, et al. Switching From TDF to TAF in HIV/HBV-Coinfected Individuals With Renal Dysfunction-A Prospective Cohort Study. J Acquir Immune Defic Syndr 2020; 85:227–232. In this study (also conducted in the Swiss HIV Cohort Study), patients with HIV/HBV coinfection also experienced improvements in ALT after switching from TDF to TAF.
6. Table 1: It is not immediately clear what “baseline” means. Authors define it in the text, and therefore there is no ambiguity there, but if readers only look at figures/tables, this might be problematic. Consider replacing “baseline” with “At TDF start” in graphs/table 1/abstract or give definitions in the captions.
7. The figures are nice and provide a clear story. However, figures 2-4 could be combined as one 3-panel figure to reduce the number of display items. In addition, consider reporting the exact p-value instead of significance stars.
8. Consider adding a conflicts of interest / funding section for transparency.
9. Line 61: should be “disoproxil”, not “dipovoxil”

Experimental design

The research question is well defined and relevant. However, I have some concerns:
1. For the outcomes, the time points must be made clearer. In lines 88-89, it should be made clear that the outcomes (viral suppression, ALT normalization etc). were assessed at TDF start, at switch from TDF to TAF, and after 48 weeks of TAF. In addition, authors should also indicate whether viral suppression was assessed at one time point, or over the whole period (baseline to switch, switch to end).
2. What definition was used to categorize patients as having chronic HBV infection? In addition, what were the indications to start treatment at your hospital? Do you follow EASL / AASLD guidelines or other? For instance, 1.2% of individuals did not have replicating HBV DNA at TDF start. On which criteria did you start NA-treatment in those individuals?
3. Authors used the MDRD formula to calculate eGFR. Since CKD-EPI may be more adequate to estimate renal disease (Matsushita et al. JAMA 2012), please explain why you used MDRD and/or consider using CKD-EPI instead.

Validity of the findings

I thank the authors for providing the underlying data. The conclusions are well balanced and supported by the study results. Although authors have used adequate statistical methods, I see one major additional limitation of the study:
1. All analyses are descriptive and the methods used are unable to adjust for major covariates (e.g. diabetes, cardiovascular disease, malignancy) which may influence changes in renal function or weight. This limitation and its potential impact (including direction) should be clearly stated in the discussion section.

Additional comments

In this study, Su et al. assessed the impact of switching from TDF to TAF among 167 patients with chronic hepatitis B (CHB) in Taiwan. TAF replaced TDF in those individuals because of a programmatic change within the hospital. Authors found that switching from TDF to TAF led to high HBV viral suppression rates, stable ALT values and increases in eGFR and body weight.

The manuscript addresses an important and timely topic and is a welcome addition to the literature. It is well-structured and nicely written, however, the methods sections need improvements in order to replicate the study, and the reporting of the results and abstract need improvements for clarity.

Reviewer 2 ·

Basic reporting

The manuscript by Su et al. is fluent to read and structured. The results are nicely put into context with background and existing data. The conclusions drawn from the data are well comprehensible. I have a only few minor comments:

1. The high percentage of patients with cirrhosis is striking - I would appreciate a list of the number of patients with cirrhosis by diagnostic criteria used (n patients with cirrhosis in ultrasound, n patients with cirrhosis in CT, etc.).
2. In line 166 it should read "the decrease in ALT".
3. In Table 1:what do you mean by malignancy? Please provide more information for the reader
4. Platelet counts, INR/Quick, triglycerides, cholesterol and alkaline phosphatase levels should be added in table 1 if available

Experimental design

no comment

Validity of the findings

no comment

---

## Round 0.2 · accepted · Accept

Thanks for addressing the minor revisions requested. Now your manuscript is accepted in PeerJ.